

# Tibetan red deer (*Cervus canadensis wallichi*) diet composition patterns and associations during lean seasons in Tibet, China

Xiaoping Liang[1], Kaili Wei[2], Qinfang Li[1], Aaron Gooley[3], Minghai Zhang[1], Jingjing Yu[1,4], Zhongbin Wang[4], Changxiao Yin[1] and Weiqi Zhang[1]

[1] College of Wildlife and Protected Area, Northeast Forestry University, Harbin, Heilongjiang Province, China
[2] Langfang Bureau of Natural Resources and Planning, Langfang, Hebei Province, China
[3] Department of Biology, Indiana State University, Terre Haute, IN, United States of America
[4] Resource and Environment College, Tibet Agricultural and Animal Husbandry University, Linzhi, Tibet Autonomous Region, China

Corresponding author
Weiqi Zhang, zhang-weiqi@nefu.edu.cn

## ABSTRACT

Tibetan red deer (*Cervus canadensis wallichi*) in the high-altitude environment of the Qinghai-Tibet Plateau could face seasonal challenges from food shortages and nutritional deficiencies but the nutritional requirements are complex. Analyzing diet composition pattern(s) is the first step to disentangle this complexity. From a systematic perspective, we hypothesize that: (A) diet composition pattern or patterns exist within the population and (B) a portion of the diet beyond characterized diet combinations will consist of random combinations. In this study, we investigated diet composition patterns of a Tibetan red deer population distributed in the Sangri Red Deer Reserve, Tibet Autonomous Region, during the harsh lean season. In March 2021 and 2022, we searched for Tibetan red deer in the reserve and collected freshly defecated samples. Diet composition at the individual level was determined using micro-histological analysis, followed by k-means clustering and co-occurrence network analysis to reveal population level diet composition patterns. Diet composition of Tibetan red deer included 14 and 19 plant species (or genera) in 2021 and 2022, respectively. K-means clustering indicated two distinct diet patterns within the population across both sampling periods. In 2021, diet composition of both clusters was dominated by *Salix* spp. (58.49% and 33.67%). In 2022, *R. macrophylla* had the highest ranking and occupied 34.83% of diet composition in the first cluster while *Salix* spp. (39.39%) was the most consumed food in the second cluster. Results of co-occurrence networks showed positively associated food combinations of less dominant food items, with a staple food occurring in all food item pairs in both years. However, randomness accounted for 95.83% and 93% of all food item pairs in 2021 and 2022, respectively, which implies a stable dietary complex system. The 2022 co-occurrence network displayed complex associations, while the 2021 network exhibited limited and simple associations. Our results suggest that Tibetan red deer fulfill their nutritional requirements by consuming high quantities of several food items or a balanced combination of foods with complex co-occurrence associations to overcome potential food shortages, but multilayer networks containing nutritional values and food availabilities are necessary to entangle the complexity of the dietary system.

## INTRODUCTION

Generalist herbivores have evolved survival strategies for high cost of reproduction habitats and their nutritional requirements can only be satisfied by either eating large quantities of food to obtain enough of the nutrient in most limited supply, or by eating a combination of food items that together fulfill the consumer's requirements (*Begon, Townsend & Harper, 2006*). Red deer (*Cervus elaphus* and *Cervus canadensis*) are adaptable generalized intermediate feeders with a wide geographic range, complex subspecies taxonomy, and diets that vary by region and habitat (*Gebert & Verheyden, 2001*; *Kay & Staines, 1981*; *Ludt et al., 2004*; *Mackiewicz et al., 2022*). Of the many red deer subspecies some are abundant and well-studied, while others are imperiled and not well known. Understanding the diets of imperiled taxa is essential to their conservation, especially in environments that are harsh, changing, or where there are non-native potential competitors.

Tibetan red deer (*Cervus canadensis wallichi*), one of the most imperiled and understudied red deer taxa, is listed as a Class I protected species in China. This subspecies was declared extinct in the wild by the World Wide Fund for Nature (WWF) in 1992, but was rediscovered near Zengqi Township, Sangri County, China (*Schaller, Liu & Wang, 1996*). This area, located north of the Yarlung Tsangpo River in the Tibet Autonomous Region, is characterized by its high altitude, harsh environment, and prolonged lean seasons that can last seven to eight months (late October to May). A subsequent field survey conducted in 2005 estimated the population to be around 220 individuals (*Shen, 2009*). After rediscovery, the population increased steadily in Sangri County for more than 30 years. However, this population may face nutritional constraints from harsh seasons and competition from sympatric livestock necessitating adoption of special foraging strategies to maintain their nutritional requirements (*Shen, 2009*; *Lv, 2020*; *Wei, 2023*; *Wei et al., 2023*). While red deer populations on the Qinghai-Tibet Plateau may be increasing, due to regional climatic trends, habitat availability is projected to decrease over 40% while habitat overlap with livestock is projected to increase 60% by the 2050s, causing increased competition for food (*Ye et al., 2023*). This highlights the conservation importance of understanding the nutritional ecology of Tibetan red deer.

Understanding the processes of nutritional ecology for large browsing ungulates that are both rare and free ranging can necessitate research on diet composition and patterns at small temporal scales, such as feeding observations or fecal analysis. Diet composition observed from a fecal sample reflects the diet composition of an individual over several hours (depending on feeding and digestion time). While each fecal sample represents a singular temporal intercept of an individual's feeding ecology, data from fecal samples is often used to determine diet composition of populations over longer time periods (*Davis & Pineda, 2016*). However, at any given sampling period, we cannot assume all individuals within a population have identical nutritional requirements or access to the same plants, which could cause different diet composition patterns in some individuals. Considering

these likely differences, dietary patterns of large ungulate populations are no doubt complex systems shaped by nutritional ecological processes. Research on dietary patterns of Tibetan red deer during the withered seasons is limited to surveys for plants with evidence of browsing at different altitudes. At low altitudes (4,100–4,300 m) evidence of browsing was found primarily on *Rosa macrophylla*, *Caragana versicolor* and *Artemisia wellbyi*, while at high altitudes (4,300–4,600 m) evidence of browsing was found primarily on *Salix* spp. and *Carex littledalei* (*Wei et al., 2023*). However, these observations likely do not capture the complexity of Tibetan red deer dietary patterns (*Wei, 2023*; *Wei et al., 2023*). Analyzing diet composition pattern(s) is the first step to disentangle this complexity. Additionally, for any complex ecological system, randomness is crucial for stability (*Barbier et al., 2021*; *Meena et al., 2023*). Therefore, from a systematic perspective, we hypothesize that (A) a diet composition pattern or patterns exist within the population (*e.g.*, similar diet composition rankings, similar preferred food items, or similar proportions of preferred food items); (B) a portion of the diet beyond characterized diet combinations will consist of random combinations.

In this study, we concentrated on collecting singular temporal intercepts (sampled feces) on multiple dates to reveal diet composition patterns and characterized diet combinations. We (i) used fresh feces to determine diet composition patterns formed by Tibetan red deer during the harsh lean season in the Sangri Protection Area and (ii) used co-occurrence network analysis to determine patterns of co-occurring food items and calculate proportion of dietary randomness. Our research will start to reveal the complexity of nutritional ecological processes from a single temporal intercept using a systematic approach, increase understanding of nutritional ecology of Tibetan red deer, and help facilitate its conservation.

## MATERIALS & METHODS

### Study area

The study area is located within the Sangri Red Deer Nature Reserve near Zengqi Village, in Sangri County, Shannan City, Tibet Autonomous Region, China (29°22′47″−29°38′10″N, 92°09′54″−92°33′11″E) (Fig. 1). The protected area is primarily characterized by high-mountain valleys, with altitude ranging from 4,000-4,900 m. The main vegetation types in this region are plateau shrublands and plateau meadows. Our study area experiences an average annual temperature of 8.2 °C and receives annual precipitation of 429 mm (*Shen, 2009*). The reserve is home to various wildlife species, including white-lipped deer (*Przewalskium albirostris*), musk deer (*Moschus chrysogaster*), snow leopards (*Panthera uncia*), and brown bears (*Ursus arctos*) (*Schaller, Liu & Wang, 1996*), as well as domestic yaks (*Bos grunniens*) and horses (*Equus caballus*).

### Sample collection

Fecal sample collection occurred using non-invasive sampling methods over a 24 day period in March of 2021 and 26 day period in March of 2022. Each day, we drove along roads in the nature reserve visually searching for Tibetan red deer. When living Tibetan red deer individuals were located, we observed them as they grazed and defecated, and once they

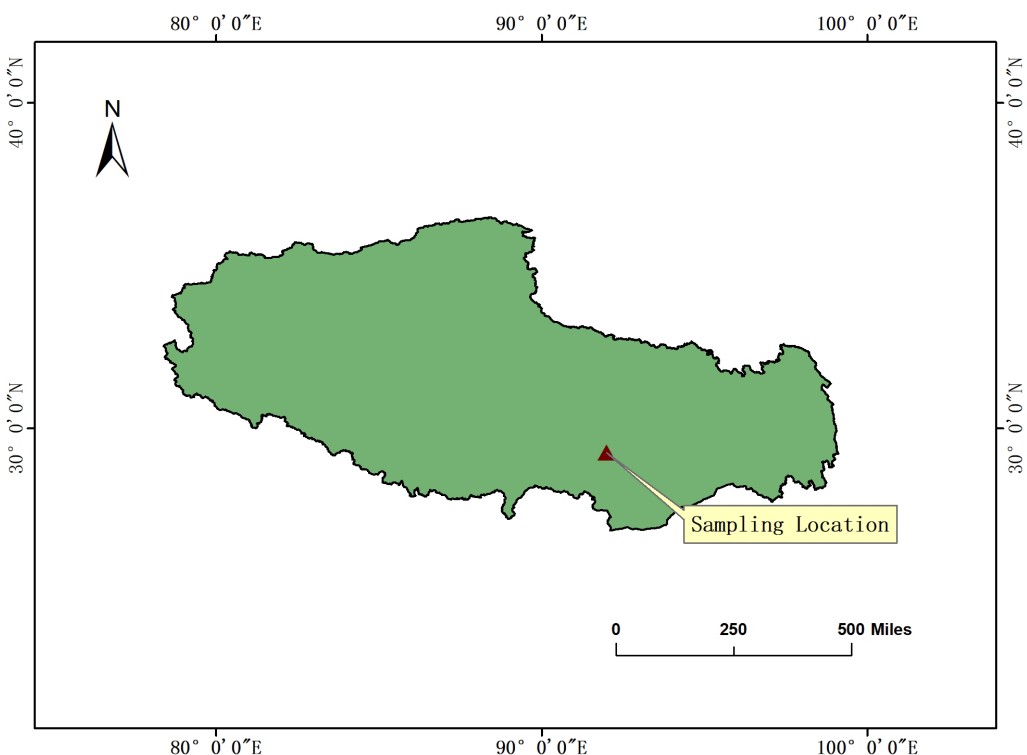

**Figure 1** **Study area.** Our sampling location is in Tibet, China.

left the area we collected the freshly defecated fecal pellets (15–20 fecal pellets from each fecal pile) that we observed them deposit and sampled vegetation. For vegetation sampling, we utilized nested plot sampling techniques to determine the minimum plot size (*Mueller & Ellenberg, 1974*) and recorded plant species within plots and collected approximately 0.5 kg of sample material for each species. We only collected samples that could be reached by Tibetan red deer (up to 2.5 m high). For grasses and forbs, entire aboveground portions of plants were collected. For shrubs, branches approximately 15–20 cm long were collected. All collected plant specimens were identified to species and stored in paper document folders prior to processing.

Diet composition was determined through fecal microhistological analysis. All plant samples and 10–15 fecal pellets from each fecal sample were oven-dried at 105 °C until they achieved a consistent weight. The plant samples were subsequently ground using a grinder and sifted through an 80–100 mesh laboratory strainer. Fragments smaller than 80 mesh and larger than 100 mesh were stored in sealed bags as experimental samples. Fecal samples were pulverized using a mortar and pestle. Separate petri dishes were used to hold reference plants and fecal sample slides. Both fecal and plant samples were individually bleached for 2-4 h in 40 mL of a 10% sodium hypochlorite solution at room temperature, after which the thoroughly digested samples were placed on a microscope slide. Samples were then rinsed with distilled water 3–4 times before the slides were sealed with glycerol and Canada balsam. Five slides were prepared for each plant species as references and ten

slides were prepared for each fecal sample. Finally, 300 plant fragments from each fecal sample were inspected under 100- or 200-times magnification and identified to the species or genus (Poaceae) level based on their epidermis and trichome morphology (*Minder, 2012*).

## Mathematical analysis

Identifiable cell wall fragments of each plant species were counted in each fecal sample and relative density (*RD*) was calculated using the formula:

$$RD = D_i / \sum D_i.$$

Where $D_i$ represents cuticle fragments of each species or genus and $\sum D_i$ represents the sum of all cuticle fragments (*Johnson, 1982*). We then compared frequencies of plant species in both fecal samples and information from plots to provide supportive information for co-occurrence analysis later. After dimension reduction of diet composition data by principal component analysis (PCA), we explored diet composition patterns of Tibetan red deer in two lean seasons within population using k-means clustering (*Celestino et al., 2018*; *Xu et al., 2015*). Optimal number of clusters was determined using average silhouette method (*MacQueen, 1967*).

In order to observe characterized food item combinations, we filtered relative densities whose values were ten times lower than the highest in each food item from all fecal samples. We then calculated relative frequency of each plant species in both fecal samples and plots. Finally, we converted diet composition data into presence-absence form and performed co-occurrence network analysis, which is based on probability models of species co-occurrence (*Veech, 2013*). Co-occurrence networks assume that co-occurrence of species is a random and independent event, and their co-occurrence is not influenced by other factors. The key concept of this model involves calculating the observed frequency ($E_{ij}$) and expected frequency ($E_{ij}$) of co-occurrence for each pair of species and comparing them to random co-occurrence. Co-occurrence probability ($P_{ij}$) represents the probability of observing the actual co-occurrence frequency or more extreme co-occurrence frequencies under random conditions.

$$P_{ij} = 1 - P\left(X \leq O_{ij}\right).$$

Where $X$ is a random variable following the hypergeometric distribution (*Johnson, Kotz & Balakrishnan, 1997*), representing the number of times species $i$ and $j$ coexist simultaneously under random conditions, and $P(X \leq O_{ij})$ is the cumulative distribution function of the hypergeometric distribution. All statistical analyses were performed in R (*R Core Team, 2024*). Package "NbClust" and "factoextra" were used to obtain optimal cluster numbers, run the machine learning based k-means clusters, and for model validation. Package "cooccur", "igraph", and "visNetwork" were used to perform and visualize co-occurrence networks.

## RESULTS

### Diet composition

Based on our observations, Tibetan red deer gathered in small groups usually consisting of three to seven individuals, sometimes more, in the lean seasons of March 2021 and March 2022. During the sampling periods a total of 39 and 50 fresh fecal samples were collected, respectively, within the boundary of the Sangri Tibetan Red Deer Nature Reserve in Sangri County. We examined 26,700 plant tissue fragments from fecal samples and 26,106 fragments were successfully identified. In March 2021, Tibetan red deer consumed a diverse array of 16 plant species or genera from 12 families. Among these, *Salix spp.* (46.53%), *Rhododendron fragariiflorum* (17.79%), and *Juniperus pingii* (13.21%) were the primary food items, collectively accounting for 77.53% of their diet. *Poaceae* (4.73%), *Dasiphora parvifolia* (2.66%), *Koenigia tortuosa* (2.58%), *Berberis temolaica* (2.37%) *Spiraea alpina* (2.12%) were less dominant food items. *Artemisia wellbyi, Ligularia rumicifolia, Anemone rivularis, Ceratostigma ulicinum, Rosa macrophylla, Carex littledalei, Betula costata, Sibiraea angustata* were occasional consumed food items and occupied less than 2% of diet composition each and less than 10% in total. In March 2022, Tibetan red deer consumed 19 species or genera from 14 plant families. *Salix spp.* (36.70%), *J. pingii* (13.53%), and *R. fragariflorum* (10.11%) remained the primary food items consumed, accounting for 60.34% of their diet collectively. Consumption of *Hippophae tibetana, Caragana versicolor, C. littledalei, B. temolaica, R. macrophylla,* Poaceae, *K. tortuosa,* and *Dasiphora parvifolia* accounted for 2%–5% of diet composition each and 30.75% combined. Occasional consumption of plants including *B. costata, S. angustata, L. rumicifolia, C. ulicinum, S. alpina. A. wellbyi, A. rivularis* and *Rhododendron nivale* comprised the rest of the diet (Fig. 2). Diet composition of Tibetan red deer exhibited consistency in the most dominant food items: *Salix spp., R. fragariflorum,* and *J. pingii* were the most frequently consumed plant species. The population exhibits strong selectivity for shrubs and forbs, which together constitute most of the diet composition.

Comparative analysis focusing on the relative frequency of various plant species present in feces and plots over two consecutive years reveals a notable trend: the species that are most prevalent within the plots do not necessarily align with those that dominate the diet composition (Figs. 3A and 3B). Specifically, Poaceae, *Salix* spp., and *D. parvifolia* emerged as the most commonly occurring species within the plots in both years. In 2021, *Salix* spp. and *R. fragariflorum* were identified in every fecal sample collected, indicating their significant dietary presence (Fig. 3A). The following year, 2022, marked a shift with *C. versicolor* being the sole species to appear in all fecal samples, suggesting a change in dietary preference or availability (Fig. 3B).

Comparisons between food item proportions demonstrate that there are outliers, in the distribution of all food items within the collected samples for each year under study. *Salix* spp. is the most dominant food item across most of the samples, with the median proportion being around 40%. However, a small number of samples exhibited a remarkably low proportion of Salix spp. in 2022, with the outliers constituting about 1–2% of the total food composition. In 2022, *R. macrophylla* accounted for a relatively minor portion of the

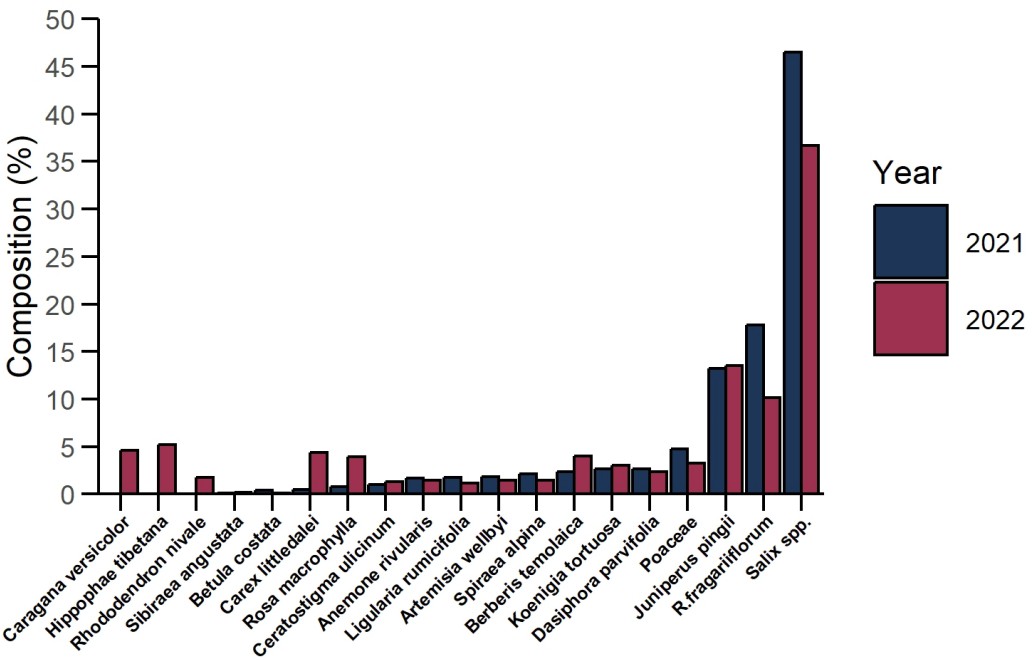

**Figure 2** Diet composition of Tibetan red deer (*Cervus canadensis wallichi*) during the withered season at the Sangri Red Deer Nature Reserve in 2021 (blue) and 2022 (red).

overall food composition, typically less than 4%, and there was a contribution from just four fecal samples, where the proportion was more than 20% (Fig. 4).

## Diet patterns

Results of PCA followed by k-means clustering showed that the food composition of Tibetan red deer during the lean seasons in both 2021 and 2022 could be clustered into two clearly distinguishable patterns. In 2021, the clustering outcomes unveiled two groups comprising 16 and 23 samples, respectively (Fig. 5A). Diet composition of both clusters was dominated in order of prominence by *Salix* spp., *R. fragariiflorum*, and *J. pingii*, but their proportions varied greatly. The first cluster consumed *Salix* spp. up to 58.49%, which was much higher than the second cluster (33.67%). In 2022, there were also two clusters, consisting of 46 samples and four samples, respectively (Figs. 6A and 6B). Unlike all other clusters, in the four samples cluster, *R. macrophylla* had the highest ranking and occupied 34.83% of diet composition. In the second cluster from 2022, *Salix* spp. (39.39%) was the most consumed food item while *J. pingii* and *R. fragariiflorum* ranked third and second within cluster, respectively, which was different from 2021.

Results of silhouette coefficients for validation demonstrated that the silhouette coefficients for the food composition in 2021 and 2022 were 0.35 and 0.64, respectively (Figs. 4B and 5B). These silhouette coefficients indicated that a substantial portion of the samples were effectively grouped into two distinct clusters. However, two samples were not similar to either of the two clusters (Fig. 5B).

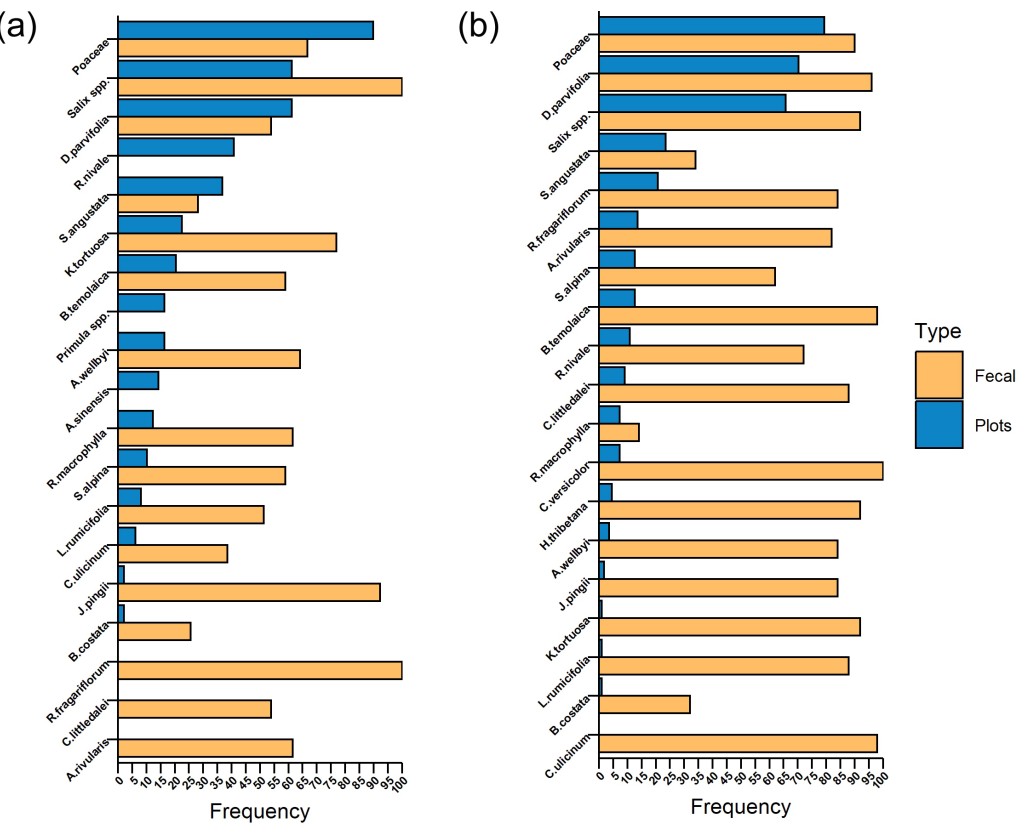

**Figure 3  Comparison of relative frequency of plant species between Tibetan red deer (*Cervus canadensis wallichi*) feces and vegetation plots during the withered season in 2021 (A) and 2022 (B).**

## Co-occurrence network of food items

Co-occurrence network analysis showed only positive associations in 2021 and both positive and negative associations in 2022. In the network, nodes had no associations with the other means and their associations were random. In the heatmap, species were removed if all associations to the other species were random (Fig. 7B). The co-occurrence network and heatmap of food items in 2021 (Figs. 7A and 7B) showed that 10 food items had nonrandom associations with at least one of the other food items. In 2021, three shrub food items including (*D. parvifolia*, *B. temolaica*, *R. macrophylla*) had positive associations with forb species, including *L. rumicifolia*, *A.rivularis* and *B. costata*. Poaceae was positively connected with *C. littledalei*. Each of these species except Poaceae occupied less than 2% of total food composition, and no associations with any of the three most dominant food items were found in 2021 (Figs. 7A and 7B).

Compared to 2021, the co-occurrence network and heatmap of food items in 2022 (Figs. 8A and 8B) showed higher complexity. Thirteen food items had positive or negative associations with other food items and most associations were among shrub species. The only association between herbaceous plants in 2022 was a positive association between Poaceae and *L. rumicifolia*. Among all food items, *Salix* spp. and *R. fragariiflorum* were two

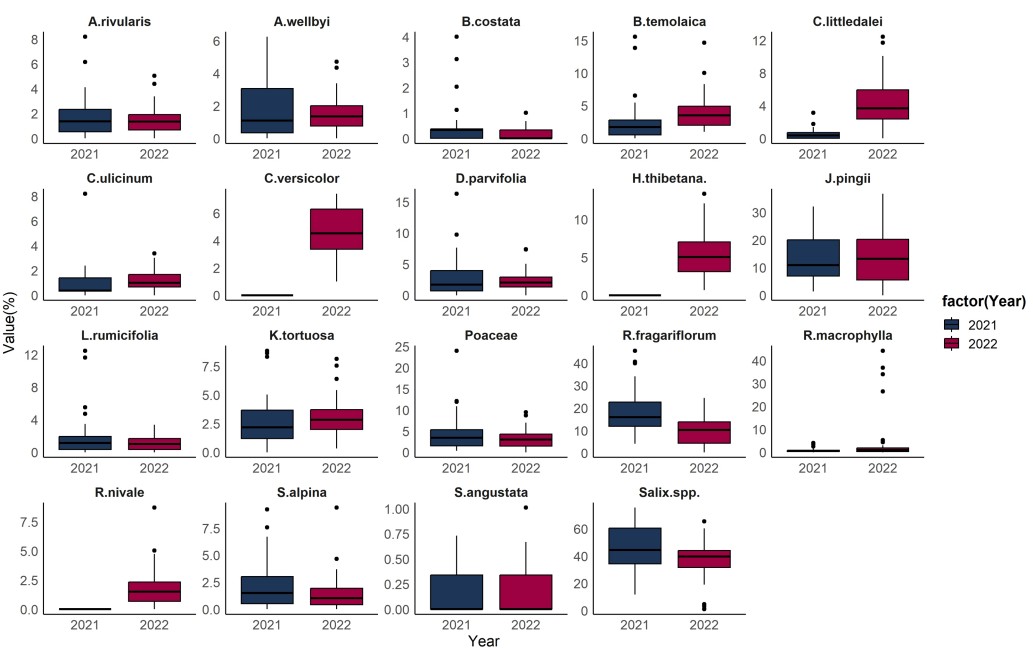

**Figure 4** Boxplot of Tibetan red deer (*Cervus canadensis wallichi*) diet composition during the withered season in March 2021 (blue) and 2022 (red).

of the three most dominant food items in cluster 1 ("Diet patterns"). *R. macrophylla* had negative associations with three shrub species (*D. parvifolia*, *Salix* spp., and *R. fragariiflorum*), suggesting that when Tibetan red deer fed on *R. fragariiflorum,* they did not simultaneously feed on those species. *Salix* spp. had positive associations with *R. fragariiflorum* (>10% of total diet composition) and *S. alpina*, which occurred in 84% and 62% of fecal samples, respectively (Fig. 3B).

The percentage of total pairings for each species showed that when considering all food item (or species) pairs, randomness occupied 95.83% (Fig. 9A) and 93% (Fig. 9B) in 2021 and 2022 respectively. Tibetan red deer did not show many consistent food item combination patterns in 2021, except for 6 food items (Fig. 9A). *Salix* spp. had a relative occurrence frequency of 100% in 2021 (Fig. 3A) meaning it co-occurred with all other food items in all fecal samples. Compared to 2021, no food item co-occurred with all other food items, and the randomness reduced by 2.83% in 2022, but still a high proportion for the network (Fig. 9B).

## DISCUSSION

Prolonged and harsh climate conditions during lean seasons on the Qinghai-Tibet Plateau potentially lead Tibetan red deer to suffer from limited food resources and decreased forage quality causing nutritional constraints. We identified diet composition patterns during the lean seasons in both of our study years that indicate Tibetan red deer rely on *Salix spp*., *J. pingii*, and *R. fragariflorum* as staple food to meet their basic nutritional needs. We infer that Tibetan red deer satisfy most nutritional requirements by eating large quantities of the
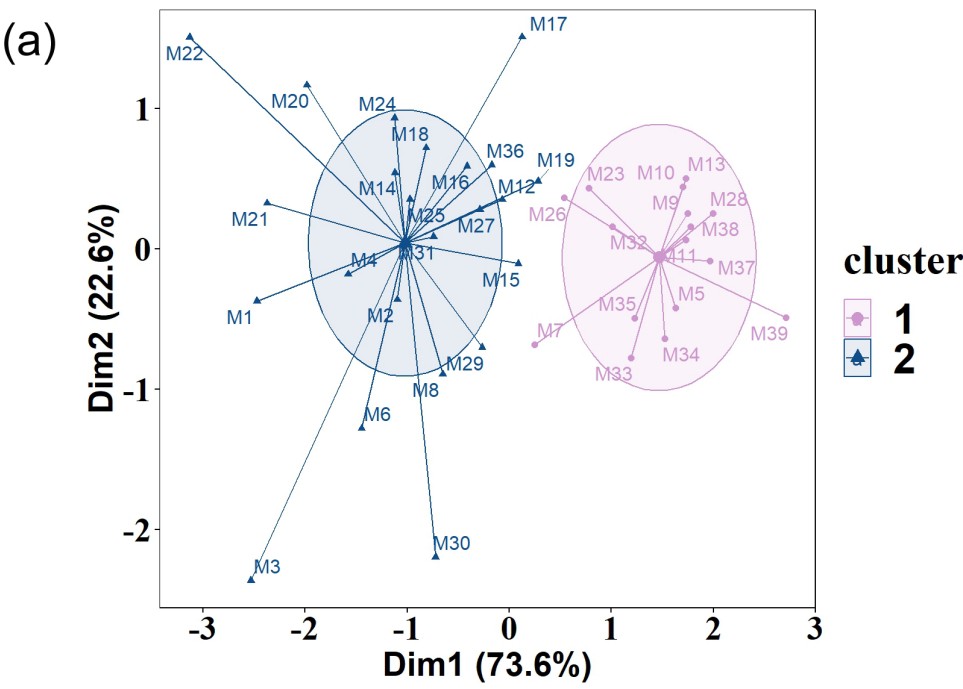

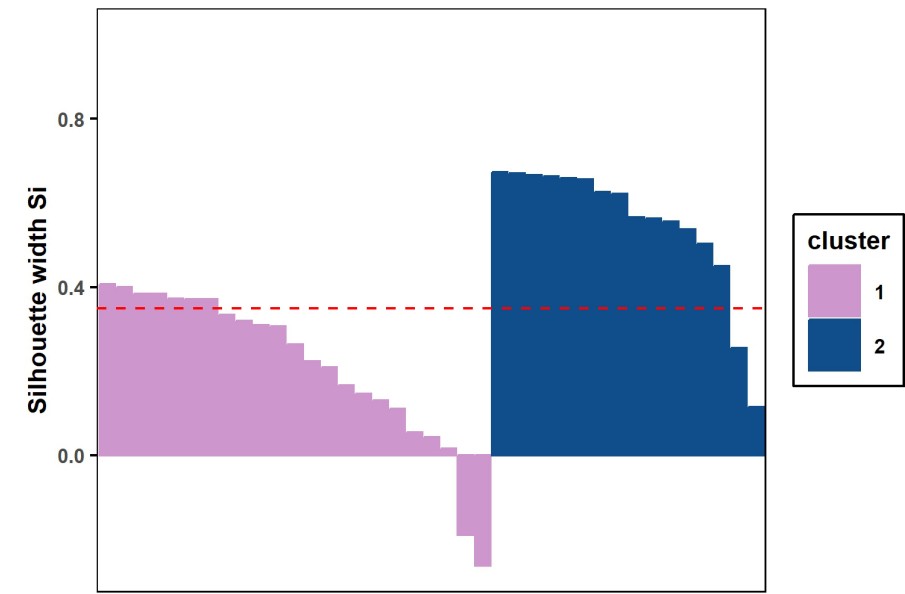

**Figure 5** (A) **K-means clustering plot of Tibetan red deer (*Cervus canadensis wallichi*) food composition in fecal pellets in March 2021.** (B) Silhouette validations for k-means cluster plots in 2021. The samples were clustered into two groups (blue triangles and purple circles).

(a)

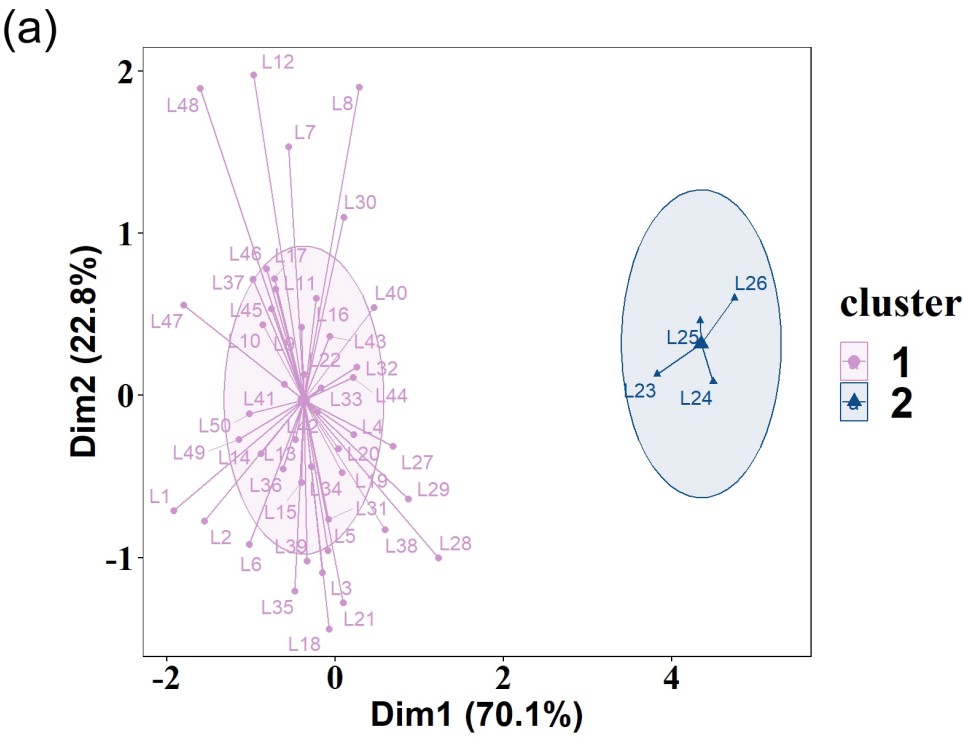

(b)

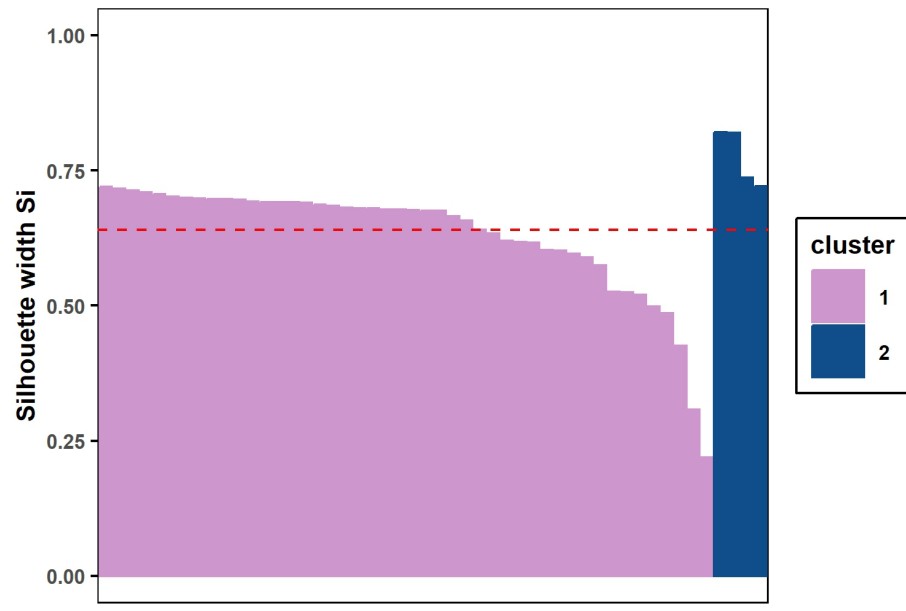

**Figure 6** **(A) K-means clustering plot of Tibetan red deer (*Cervus canadensis wallichi*) food composition in fecal pellets in March 2022.** (B) Silhouette validations for k-means cluster in 2022. The samples were clustered into two groups (blue triangles and purple circles).

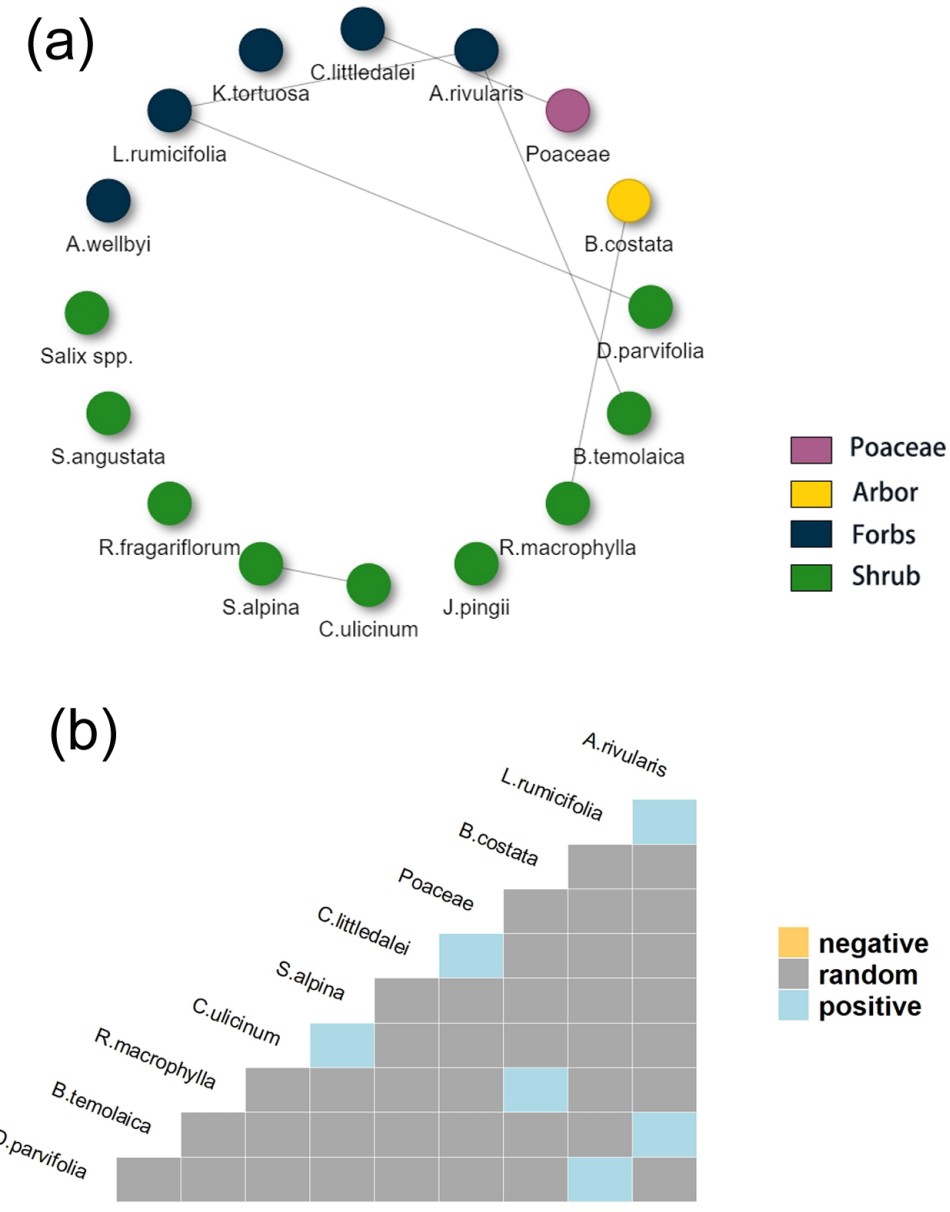

**Figure 7** (A) Co-occurrence network of food items in Tibetan red deer (*Cervus canadensis wallichi*) fecal in March 2021. (B) Heat map showing the positive species association. (A) Edges between nodes represent significant co-occurrence relationships among plant species in 2021, solid lines represent positive associations. (B) Species names are positioned to indicate the columns and rows that represent their pairwise relationships with other species in 2021.

three top ranking food items. However, a boxplot for both years showed most food items exhibited several outliers in composition. For example, a high proportion of *R. macrophylla* and low proportion of *Salix* spp. This phenomenon indicated that one or more individuals within the population exhibited distinct diet preferences and did not rely solely on staple food species to meet their nutritional requirements. This suggested that even at short

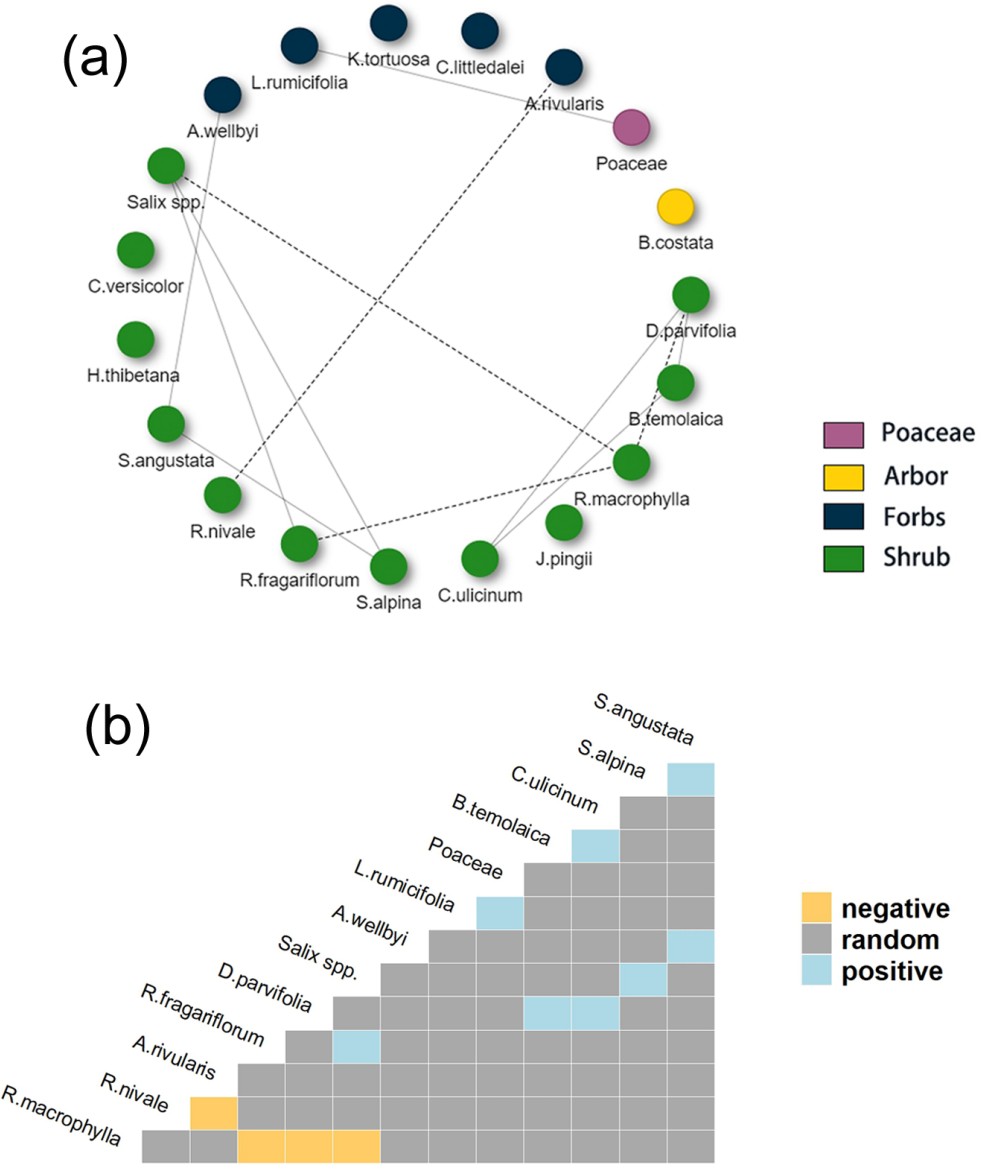

**Figure 8** **(A) Co-occurrence network of food items in Tibetan red deer (*Cervus canadensis wallichi*) fecal pellet in March 2022. (B) Heat map showing the positive and negative species associations.** (A) Edges between nodes represent significant co-occurrence relationships among plant species in 2022; (B) solid lines represent positive associations and dashed line represent negative associations. Species names are positioned to indicate the columns and rows that represent their pairwise relationships with other species in 2022.

temporal scales of 2–3 weeks, there are hidden diet composition patterns at the individual level and within the Tibetan red deer population.

During the fieldwork conducted in 2021, there were no *R. fragariflorum*, *C. ulicinum*, and *A. rivularis* collected within the plots, highlighting a potential discrepancy between

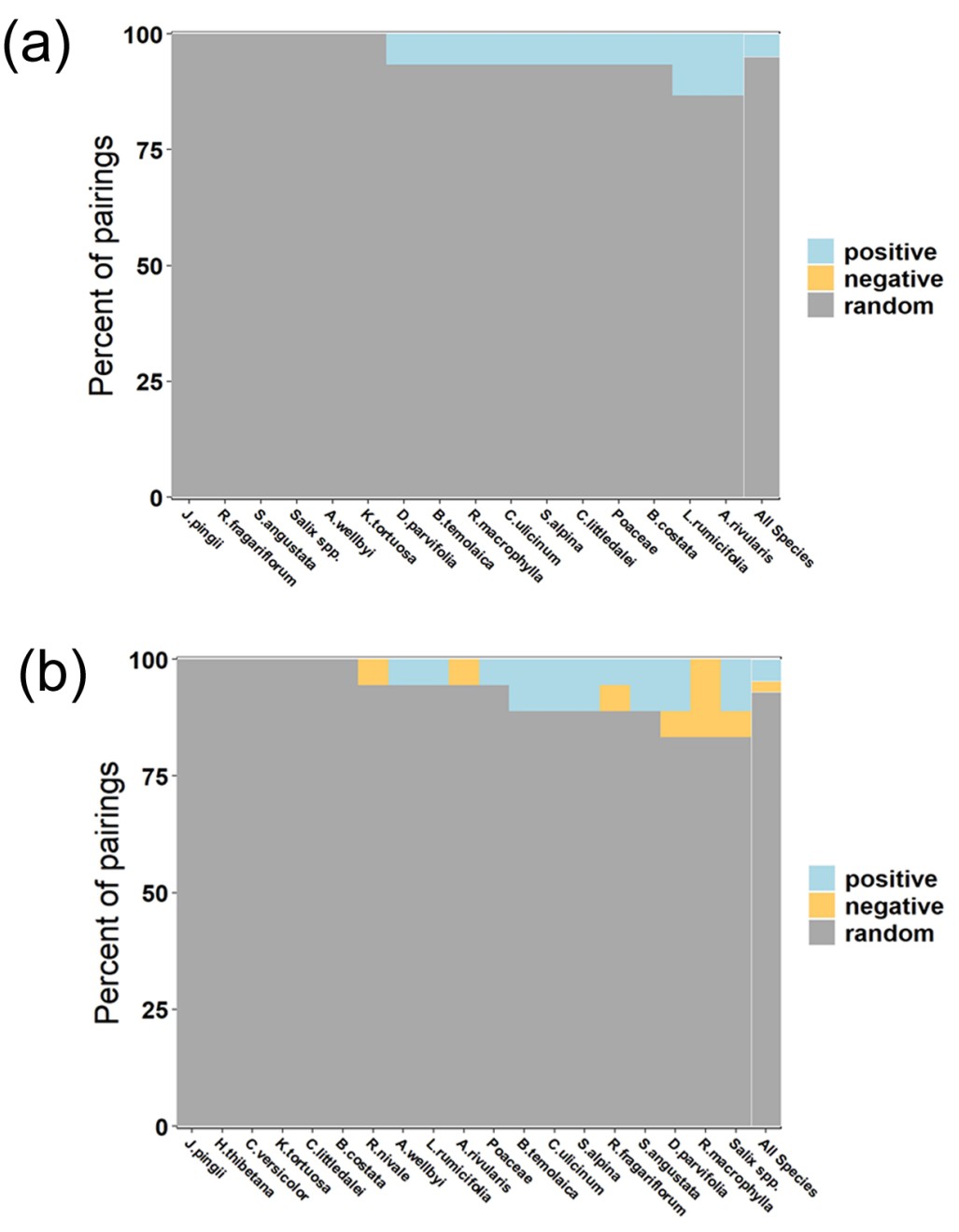

**Figure 9** Boxplot showing the percent of total pairings for each species found in Tibetan red deer fecal pellets that are positive, random, or negative in March 2021 (A) and 2022 (B). Species are ordered by increasing number of total associations. The right-most bar, outlined in white, represents the assemblage-wide percentages.

the observed diet and the available flora. To address this gap and enhance the accuracy of species identification, plant samples were collected in 2022 as a supplementary effort. This additional collection aimed to provide a more comprehensive understanding of the species composition and its variations over the years, thereby enriching the dataset for

more accurate comparisons and analyses. Comparisons between food item proportions demonstrate exceptional cases, or outliers, in the distribution of all food items within the collected samples for each year under study. These outliers represent proportions that are different from the majority of diet composition. *Salix* spp. was the most dominant food item across most of the samples, indicating that, on average, nearly half of the food composition in all samples was made up of *Salix* spp. However, a small number of samples exhibited a remarkably low proportion of Salix spp. in 2022. This substantial deviation from the norm suggests that in these specific instances, *Salix* spp. was consumed much less than usual. Additionally, in 2022, *R. macrophylla* accounted for a relatively minor portion of the overall food composition indicating that in these samples, *R. macrophylla* was consumed in much higher quantities than what was generally observed, highlighting a possible preference or increased availability of this food item in certain situations. The analysis further revealed that, with the exception of *J. pingii* and *C. versicolor*, all other food items in the study exhibited outlier proportions. This suggests that the consumption of these two food items was relatively stable across the samples, without the extreme variations in proportion that characterized the other food items (Fig. 4). Though, to some extent arbitrary, two distinct diet composition patterns have emerged during the lean seasons in both years. Therefore, we can infer that Tibetan red deer have diet composition preferences, but they vary within the population as well as in proportional amounts. Samples within clusters have similar intake amounts of each dietary component and display preferences to certain food items. Two samples did not group with the two clusters in 2021, suggesting the existence of additional diet composition patterns. A high proportion of *R. macrophylla* in four samples in 2022 suggests that *R. macrophylla* was a staple food of some individual(s), but not others. Regardless of whether the four samples were from the same individual, we can infer that at times some individual(s) have relatively different diet compositions and probably different nutritional intake patterns from those who take *Salix* spp. and *R. fragariiflorum* as staple food. It is likely that not all individuals have the same nutritional requirements at any given time, and each may adjust their dietary intake accordingly.

Causes of the different diet patterns could be various, including food resources availability, intra- and interspecific competition, nutritional content (*Agetsuma et al., 2019*; *Hobbs et al., 1991*; *Murray et al., 2016*), habitat differences (*Araujo, Bolnick & Layman, 2011*; *Bolnick et al., 2003*; *Proulx, Promislow & Phillips, 2005*), and herbivore-plant interactions (*Perkovich & Ward, 2022*). Vegetation types in the Sangri protected area are primarily composed of alpine shrub meadows where the staple foods of the Tibetan red deer are relatively abundant (*Hu, 2002*; *Lv et al., 2020*). Grazing livestock (yaks and horses) that primarily consume herbaceous plants from the Cyperaceae and Poaceae families (*Lv, 2020*) are present on the Sangri Red Deer Nature Reserve. Although Tibetan red deer are typical mixed browsers, grazing livestock could compete with them for certain plants (*Ye et al., 2023*) or hinder access to certain feeding areas. In other areas, dietary overlap between deer and cattle (*Bos taurus*) may increase when forage availability decreases during the winter (*Ortega et al., 1997*; *Thill & Martin, 1989*) and there is evidence that livestock can alter the feeding behavior of deer or displace deer from foraging areas (*Chaikina & Ruckstuhl, 2006*; *Stewart et al., 2002*; *Weiss et al., 2022*). More investigation is needed

to determine if free-ranging livestock impact the foraging behavior of Tibetan red deer. Different dietary composition patterns were observed in a red deer population inhabiting a typical forest-grassland steppe without livestock in the Inner Mongolia Gaogesitai Hanhula National Nature Reserve. The population foraged mainly on deciduous trees and shrubs, such as *Quercus mongolica*, *Armeniaca sibirica*, *Salix rosmarinifolia*, *Ulmus pumila*, and a relatively high proportion of Poaceae. However, a high proportion of *Picea asperata* was observed in some samples, which is considered an indicator of food shortages in winter (*Zhou et al., 2022*).

Ungulates regulate selection and consumption patterns of plant species with limited availability and form diet combinations of available vegetation (*Provenza et al., 2003*; *Ortíz-Domínguez et al., 2022*). When characterizing diet combinations from co-occurrence networks, lack of associations does not indicate lack of co-occurrences. In 2021, though the network did not reveal associations for *Salix* spp., *Juniperus pingii*, and *Rhododendron fragariiflorum*, this did not indicate lack of co-occurrence with any other plant species. On the contrary, they occurred in almost all fecal samples which made them appear to have no co-occurrence associations and appear random. Positively connected food items appeared to play a role of complementing unsatisfied nutritional requirements, as all positively connected food item accounted for less than 5% of diet composition each (*Hebblewhite & Pletscher, 2002*). The co-occurrence network in 2022 exhibited higher complexity. It was interesting that *R. macrophylla* accounted for less than 5% of diet composition but was the most dominant food item in cluster 2 and had negative associations with another three shrub species (*D. parvifolia*, *Salix* spp. and *R. fragariflorum*). Two of them (*Salix* spp. and *R. fragariflorum*) are considered as staple food in the other three clusters in both 2021 and 2022. This suggests that such diet combinations might meet a different nutritional requirement of some deer or are caused by other ecological associations.

As we hypothesized, a large proportion of the diet consisted of random combinations in the co-occurrence network. Positively co-occurring combinations make up only a small fraction of all possible combinations. The presence of this randomness reflects the diversity and adaptability of the Tibetan red deer's diet in a variable resource environment and should not be overlooked. Such dietary diversity and adaptability could help Tibetan red deer populations survive projected habitat changes (*i.e., Ye et al., 2023*). Additionally, characterized diet combinations and patterns of ungulate browsing in small temporal scales can be influenced by community composition and structure.

Proxies for determining an animal's diet can represent different temporal periods and scales, and sometimes yield different information (*Davis & Pineda, 2016*). Our study identified somewhat different dietary compositions compared to field surveys of plants with evidence of browsing (*Wei et al., 2023*). The surveys identified *Salix* spp., *Rosa macrophylla*, and *Dasiphora parvifolia* as comprising >50% of Tibetan red deer diet, whereas our fecal analysis identified *Salix* spp., *R. fragariiflorum*, and *J. pingii* as the primary components of diet. Furthermore, the browsing evidence surveys showed that diet varied with altitude and plant community, however other than *Salix* spp., plants identified as primary diet components were only minor components of the diet composition we identified from feces. It is possible that feeding on *R. fragariiflorum* and *J. pingii* occurred at other locations

or evidence of feeding on these species was not as readily identifiable as on other species. Deposited feces might not contain forage eaten at the location of deposition and could be composed of forage from multiple locations (*Picard et al., 2015*). This highlights why determining diet for large highly mobile browsing ungulates can be complex and benefit from multiple methods of investigation.

Our co-occurrence network analysis revealed dietary associations in the feces of Tibetan red deer. High proportion of randomness in both years implies a relatively stable dietary system that would help the population overcome potential nutritional deficiency during the lean season. However, more information is needed about the nutrient contents of dietary items, as well as how plant community structure is related to food availability and dietary choices of Tibetan red deer to better understand this complex network.

## CONCLUSIONS

In lean seasons, Tibetan red deer predominantly feed on *Salix* spp., *J. pingii*, and *R. fragariflorum* in most circumstances. More attention should be given to diet compositions that have outliers, as they might imply different nutritional requirements or environmental changes in a small temporal scale. Diet composition patterns within the population exhibit variability in proportion and rankings of dominant food items across k-means clustering of food composition. Positive associations in co-occurrence networks suggest Tibetan red deer select some food combinations to complement unsatisfied nutritional requirements in addition to consuming staple foods. Negative associations represent different diet composition patterns during the lean season, but the reasons for such associations are unknown and need further investigation. Additionally, high randomness between many dietary item co-occurrences across individual and population levels should not be ignored as it might represent adaptation to a complex and changing environment.

## ACKNOWLEDGEMENTS

We appreciate the help of deputy director Suolang of Sangri County Forestry and Grass Bureau, as well as Sangzhu and Ouzhu for guiding us during field work.

### Funding

This work was supported by the National Natural Science Foundation of China (No. 32071512, 31500328) and Postdoctoral Startup Funding of Heilongjiang Province. The funders had no role in study design, data collection and analysis, decision to publish, or preparation of the manuscript.

### Grant Disclosures

The following grant information was disclosed by the authors:
National Natural Science Foundation of China: 32071512, 31500328.
Postdoctoral Startup Funding of Heilongjiang Province.

## Competing Interests

The authors declare there are no competing interests.

## Author Contributions

- Xiaoping Liang performed the experiments, analyzed the data, prepared figures and/or tables, authored or reviewed drafts of the article, and approved the final draft.
- Kaili Wei performed the experiments, prepared figures and/or tables, and approved the final draft.
- Qinfang Li performed the experiments, prepared figures and/or tables, and approved the final draft.
- Aaron Gooley analyzed the data, authored or reviewed drafts of the article, and approved the final draft.
- Minghai Zhang conceived and designed the experiments, authored or reviewed drafts of the article, and approved the final draft.
- Jingjing Yu performed the experiments, prepared figures and/or tables, and approved the final draft.
- Zhongbin Wang performed the experiments, prepared figures and/or tables, and approved the final draft.
- Changxiao Yin analyzed the data, prepared figures and/or tables, and approved the final draft.
- Weiqi Zhang conceived and designed the experiments, analyzed the data, prepared figures and/or tables, authored or reviewed drafts of the article, and approved the final draft.

## Data Availability

The raw data is available in the Supplementary Files.

## Supplemental Information

Supplemental information for this article can be found online at http://dx.doi.org/10.7717/peerj.18614#supplemental-information.

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
