# Peer review of "Tibetan red deer (*Cervus canadensis wallichi*) diet composition patterns and associations during lean seasons in Tibet, China"

_PeerJ, doi:10.7717/peerj.18614_

## Round 0.1 · original submission · Major Revisions

Thank you very much for your manuscript titled “Tibetan red deer (Cervus elaphus wallichii) diet composition patterns and associations during withered seasons in Tibet, China” that you sent to PeerJ.

This study presents very valuable descriptive information for the knowledge information on red deer present in a region of Tibet, however, there are some weak points that need to be clarified.. For example, one reviewer considers that there is a poor use of the available literature, which causes an introduction that lacks basic references to the current knowledge of the diet in the species and an absence of a conceptual framework appropriate to the study problem. On the other hand, the study lacks research questions and hypotheses; in addition, the statistical approach is not well explained or justified and some conclusions are too general and are not linked to the results, which causes the lack of solid conclusions. One reviewer considers that a genetic evaluation of the feces is necessary to avoid confusion in its identity.

Please note that we consider these revisions to be important and your revised manuscript will likely need to be revised again.

·

Basic reporting

The manuscript entitled "Tibetan red deer (Cervus elaphus wallichii) diet composition patterns and associations during withered seasons in Tibet, China" explores the dietary adaptation of this scarcely studied subspecies of red deer. As later described, there are important issues about the correct use of the available literature which do not allow to contextualise the study correctly. The dataset is interesting, but it is not well used, as later explained.

Experimental design

The weak introduction does not allow the formulation of relevant questions or clearly detect research gaps that the study aims to fulfil. These are indeed missing from the introduction, which, apparently, just aims “To further understanding of the nutritional ecology of Tibetan red deer during the harsh withered season…” (L83-84). The methods section needs references, and the selected statistical approach also needs justification. It is not clearly linked to the (missing) research questions and it is not clear what is expected from them.

Validity of the findings

The impact and novelty of the study are low. The study applies standard techniques to a new species and yields somehow expected results, which definitely are of interest for scientists working with the subspecies or diet selection in general but do not bring any new methodological or theoretical advances. The statistic is not well justified and, considering the gaps in the introduction, I do not feel that the conclusions are well supported.

Additional comments

“Generalized consumer” (L45) is not a common term in nutritional ecology. “Generalist species” should be used instead.
The authors comment on the complex taxonomy of the species, citing Lundt et al. 2004. Indeed, this work concludes that wallichii belongs to the canadensis group and thus, its correct scientific name is Cervus canadensis wallichi (nomenclature accepted by the IUCN; out of discussion as far as I know). The authors should thoroughly read the papers that they cite, so they can be cited properly.
Correct the typo error in L54.
The statements in L56-58 are supported by the references Wei 2023 and Wei et al. 2023. The first is an inaccessible Master thesis, which seems based on the same data presented in this manuscript. That looks like circular reasoning: using a similar work previously published as an MSc thesis for supporting the current, probably same statements. Please, avoid this. About the second reference (by the way, it appears in the references as We et al. 2023 – correct the typo error), it must be clarified. According to the abstract, it corresponds to the same fieldwork and is written by the same authors, which suggests that the data presented is the same as in this study, probably just with a slightly different scope. Wei et al. 2023 seem focused on the plant availability in the study area but also provide diet selection data. This raises important issues. The reference is (properly) used just once in the discussion (L323-333) and seems to reach quite different results and conclusions compared to those in this study. However, in the introduction the reference is just used for commenting that “research on the dietary patterns of Tibetan red deer during the withered seasons is limited” (L59-60). The authors must expand the key findings of this previous study in the introduction, so the readers have an adequate background about what is already known. Indeed, it is quite surprising that the results from this study are omitted from the introduction.
Similarly, the reference Shen, 2009 (another inaccessible master thesis entitled “Food composition, feeding and nutritional adaption strategies of Tibetan red deer (Cervus elaphus wallichi) during the green grass period in Tibet” is just used to provide data on the population size but not about diet and nutritional adaptations of the subspecies, which obviously is much more relevant information for understanding this study. The conclusion of this and the previous comment are clear: the authors are not adequately using the available literature to provide all the necessary basic information to the readers.
L85. The authors can not guarantee that the samples correspond to multiple individuals. Please, correct the sentences according to this.
L101. Correct “caballas”.
Samples collection – How can the authors ensure that the faecal samples do not correspond to sympatric white-lipped and musk deer?
L110. We only “collected” samples…
L114-127. Please, provide references for the methodology.
L125. Why 300? Most studies use 100.
L177-180. This doesn´t look like an adequate presentation. The figure seems to show the percentage of plots and faecal samples where each plant appears. Similar studies tend to compare the over percentage representation of each plant both in the faeces and the samples and compare these percentages through several selectivity indices. Feeding predominantly in one or few items while routinely sampling other plants (for example, to monitor when these can be nutritionally suitable thanks to maturation, increase in nutrients, decrease in plant secondary compounds, etc.) is a common strategy in ruminants. Thus, it is not surprising that plants with a low total appearance in the faecal samples (like those under 2%) still show a much higher frequency in the way of counting shown in these figures. The authors must reconsider this approach.
L187-209. All this is described as a discussion. Please, provide just results here and move all the interpretations to the discussion. Moreover, the information presented here fits quite well with my previous comment: strong differences in the consumption of a certain species within the two studied years are indeed common and must simply reflect phenological changes in the plants; probably, one plant was not “ready to eat” in one of the years but ready in the other one.
The “diet patterns” analyses are very unclear to me. First, the PCA is based on which variables? To which variables are connected each of these two dimensions? I would even add that there are several PCA methodologies mainly linked to the rotation method which are important for understanding the statistical approach. All this basic information to understand the figures is not provided. Second, what is represented in the plots? The samples? The plants? Further than not being well explained, what is the point of this analysis? Why is it necessary to group the samples? What relevant question do we answer with this analysis?

Reviewer 2 ·

Basic reporting

The manuscript deals with Tibetan red deer diet. The text provides interesting data, however English should be improved and a hypothesis should be added at the end of Introduction. I Suggest the sentence form page 8, starting from line 266 (low or high dietary species richness) during lean season (harsh winter).
References are OK, however red deer is an intermediate feeder, not always browser.
Abstract is wordy,largely descriptive and redundant

Experimental design

I have some concerns to methodology.
Given the fact, other ungulates are present - were Authors sure they always collected stools of the study species? From my experience it is really difficult. Genetic species (cervid) identification of stools is lacking).
Pseudoreplicates (stools form the same individuals) may affect results. Again, multilocus genotypes would could help.
I did not find calculation for dietary species richness. Was food preference analysis performed? Ivlew index was used?

Validity of the findings

New data on diet of Tibetan red deer. Since stool assessment to a deer species by genetics was not performed, I am not sure data are fully controlled.
Hypothesis is missing, thus conclusions are blurred and too general.

Additional comments

Minor comments I put in a doc file

Annotated reviews are not available for download in order to protect the identity of reviewers who chose to remain anonymous.

·

Basic reporting

The manuscript titled "Tibetan red deer (Cervus elaphus wallichii) diet composition patterns and associations during withered seasons in Tibet, China" is well-written and offers valuable insights. The study, conducted over two years using deer pellet microhistological analysis, provides interesting information about the diet of Tibetan red deer. It would be beneficial to include details on the current status of the population, such as the number of individuals and their conservation status. Additionally, the significance of conducting such studies on this subspecies and in this specific region should be emphasized more. Including a map to illustrate the uniqueness of this population would also be useful. The findings about this particular population are intriguing, and it would be even more compelling to include information on the population structure, such as the ratio of males to females and the age distribution (young and adult). If the pellet collection involved observing the individuals, this data could also enrich the paper. Since Cervus elaphus has a broad distribution, comparing the results with other populations would add further value. I found this paper very engaging, and it has sparked my curiosity. I've marked some typographical errors in the attached PDF, and I hope these comments are helpful to the authors.

Experimental design

The research methodology is thoroughly planned and clearly presented.

Validity of the findings

The results are clearly presented, with robust data and well-managed statistical analysis. The conclusions are derived from original research and are clearly articulated and effectively presented.

---

## Round 0.2 · Minor Revisions

After reviewing this revised version of your manuscript, I see that the main comments suggested by the reviewers have been included. However, there are still some details that need to be clarified before having a final version that can be published.

It is necessary that the hypothesis be even clearer, especially in points B) and C). It is also required that these points of the hypothesis be considered within the discussion of its results.

Reviewer 2 ·

Basic reporting

This is a second time, I red the ms - now the corrected version. I greatly appreciate changes in the ms s well as the explanations.
I thank Authors for adding hypotheses in the Introduction section. However, they are not clear to me (despite I am not a native speaker, I still claim English should be proofread/corrected.
In Discussion section, I only found new comment on randomness of a diet, I did not find where in Discussion Authors address to their other hypotheses.
Abstract is still descriptive, hypotheses and their significance not mentioned in Abstract.

Experimental design

Authors explained my concerns quite well.

Validity of the findings

Seems OK, however given the fact English needs profreading, sometimes I had difficulties in following the text.

Additional comments

None,

·

Basic reporting

The authors of the manuscript titled Tibetan red deer (Cervus canadensis wallichi) diet composition patterns and associations during lean seasons in Tibet, China' have addressed my comments, as well as those of the other reviewers, very effectively. They have used the feedback to strengthen their arguments throughout the manuscript. It is great that the authors have included a map, which will help readers better locate the Tibetan red deer population studied in this paper.I believe it is now ready for publication.

Experimental design

The research methodology is well-planned and clearly presented. The authors have incorporated the reviewers' recommendations, making their presentation more precise.

Validity of the findings

The results are clearly presented, with robust data and well-managed statistical analysis. The conclusions are derived from original research and are clearly articulated and effectively presented.

---

## Round 0.3 · accepted · Accept

After reviewing this revised version of your manuscript, I see that the main comments suggested by the reviewers have been included. Therefore, I am satisfied with the current version and consider it ready for publication.